# Young Children’s Housework Participation in Taiwan: Serial Multiple Mediations

**DOI:** 10.3390/ijerph192315448

**Published:** 2022-11-22

**Authors:** Ching-Fen Lee, Shain-May Tang

**Affiliations:** 1Minghsin University of Science and Technology, Hsinchu 30401, Taiwan; 2National Open University, New Taipei City 247031, Taiwan

**Keywords:** young children’s participation in housework, children’s health, parents’ attitude, child’s housework preferences, serial multiple mediation

## Abstract

The purpose of this study was to investigate not only the participating in housework but also the parents’ attitude and child’s preference, in relation to children’s health and housework participation in Taiwan. We collected data from the Young Children’s Housework Participation Questionnaire on “Google forms”. A total of 755 parents with preschool children living in Keelung City, Taipei City, and New Taipei City participated through the snowball method. The results showed that children’s health not only directly affected their housework performance but also indirectly influenced their housework participation through the serial multiple mediation of parents’ attitude and the child’s housework preference. Therefore, this study confirms that, when analyzing the factors of children’s housework participation, it is necessary to have a clearer understanding of the relationship between variables to further construct a more complete model framework that affects children’s housework participation. Additionally, it is very important for parenting education to improve parents’ attitude towards the importance of children’s housework and encourage children’s preference for housework.

## 1. Introduction

Most of the housework research has mainly focused on how husbands and wives divide their labor at home [1]. For quite a long time, research on children’s participation in household chores was restricted, most likely because it was considered of minor importance in household labor contribution [2,3]. However, recently researchers have begun to pay attention to children’s participation in housework, because many studies have found that children’s involvement in housework in childhood has many positive effects. In a longitudinal cohort study, housework participation in early childhood is associated with later development of prosocial behavior, academic ability, peer relationship, and life satisfaction [4]. Childhood chores also lead to increased competence with necessary life skills and happiness [5], decreased risk of drug and alcohol [6], and increased future housework participation in older age [2,7]. In addition, the routine of household chores plays a vital role in cultivating children’s gratitude, since children’s housework participation not only make them appreciate the hard work of their parents, but also makes the parent–child relationship better and further develops a long-term grateful disposition [8].

Cultural differences have an impact on the values of different countries and families, which, in turn, affects the performance of children’s housework ability [9]. Western society believes that it is their family responsibility for children to participate in housework, and it can also reduce the burden of housework on parents. Therefore, children are trained to do housework from an early age and are given the responsibility of doing housework [10]. Under the influence of Confucian culture in Taiwan, many parents are concerned about their children’s academic performance and believe that academic performance will have an important impact on their children’s future [11,12], so they do not think that it is important for children to participate in housework. However, modern Taiwanese society is influenced by Western values, and parents’ attitudes and expectations for their children’s housework participation have also changed [9]. Under the impact of Confucian culture and Western values, Taiwan certainly has its own characteristics, which is what this study intends to present.

A common descriptive finding of research on children’s housework participation is that housework involvement differs by the child’s gender and age [13,14]. Girls tend to spend more time in domestic work than boys [15,16,17]. Older children do more housework than younger children [18,19]. The perspective of gender-role socialization is often used to explain the difference between girls and boys through the reproduction of parents’ housework division [10,20,21]. Older children not only do more chores, they also do more complex chores than younger children. This concept is explained by developmental theory [22].

Recent research has also found that the timing at which children begin to engage in household chores matters. Rossmann [23] indicates that housework participation in early childhood has a greater impact on a child’s future success in the completion of education, getting started on a career path, and relationships with people than when it begins in adolescence. However, many past studies on children’s housework in Taiwan have focused on children over the age of 9, because these children have more mature housework skills and better reading skills than preschoolers, and it is easier to obtain relevant information through questionnaires [24]. Therefore, the extent to which young children participate in housework is not clear, which is one of the reasons why this study will analyze the participation of young children, aged 3–6 years, in housework.

In addition to the influence of gender and age factors, children’s health was the focus of this study, since this was often overlooked when discussing the division of household chores. In the past, limited studies have analyzed the participation of children with physical disabilities [25,26] and psychological problems, such as ADHD [27,28,29], in housework. Younger children with physical disabilities tend to do fewer household chores [26]. In families with ADHD children, parents and other siblings tend to assist the ADHD children with household chores [28]. However, children with ADHD can still do a lot of housework [29]. A study using survey datasets from six countries discovered no significant correlation between household chores and health [30]. The inconsistent findings also point to the need for further clarification of the relationship between children’s health and housework participation.

The socialization model is the most used to explain children’s household participation, referring to the learning expectations, behaviors, skills, values, and other obligations necessary for participation in social groups [10]. For children, the family is their earliest contact and the most influential place. Parents are not only imitators of their children, but also important others who influence their children’s values and behaviors [31]. Therefore, past research has also discussed the impact of parental education, gender role attitudes, employment, division of housework, and marital status on children’s participation in housework [2,21,32]. Parents’ attitudes toward their children’s housework participation are of concern to this study, since they not only affect their children’s housework arrangements [33], but also parental guidance [34] and encouragement [35] in chore involvement.

Another variable of interest in this study is children’s housework preferences. According to the process and activity theory, based on the concept of positive psychology, people gain happiness by participating in activities. At the same time, people are more willing to participate and are more persistent in participating in activities when they find them interesting and have sufficient skills to perform them [36]. When children think housework is fun or are interested in housework, they are not only happy about doing chores, but also more willing to share housework [37,38]. Signorielli and Lears [39] mentioned that children’s attitudes toward household chores were significantly related to the behavior of housework participation.

The factors that affect children’s participation in housework can be direct or indirect [40]. For example, both children with disabilities [27] and a parent’s attitude towards children’s participation in housework [41] can directly affect children’s housework participation, but children’s health can also alter the parental attitude, which, in turn, affects children’s housework participation [26]. Although believing housework sharing is important for young children, most parents will not expect the sick or disabled child to participate it because the participation in housework is not so important, compared to the child’s health problems [9]. When parents do not expect their sick children to do chores, they will not encourage their children to do them, resulting in less involvement in chores [34] or even less interest in chores [9]. However, if parents think that children’s housework is important, they often give children more guidance and assistance [42]. Thus, children have more confidence and interest in doing housework [43], and then participate in more housework [44]. Therefore, when we understand children’s housework participation, it is not only necessary to analyze which factors affect children’s housework, but more importantly, how the relationship between these factors affects children’s housework participation. Based on this concept and previous research results, this study would like to exam whether the serial multiple mediators of parental attitude and child preference can mediate the relations between children’s health and young children’s housework.

This study also considered children’s gender, parental division of housework, and whether the child had siblings as control variables to avoid interference from other factors that influenced our research model. Previous studies have found that children’s participation in housework varies by gender [14,45], parental division of housework [46], and the presence of older siblings [47]. 

## 2. Methods

### 2.1. Samples

A cross-sectional design with a convenience sample through Google Sheets was used for this survey study. The respondents were parents with young children attending preschools in Keelung City, Taipei City, and New Taipei City. After excluding 52 invalid respondents, a total of 775 parents were used in this study. The regional distribution of respondents was 219 in Keelung City, 366 in Taipei City, and 190 in New Taipei City. Taipei City, New Taipei City, and Keelung City are all located in northern Taiwan and have similar characteristics [48,49], such as population density, the proportion of the population with a college degree or above, and the proportion of the population aged 15 to 64 who are able to work. Additionally, they all belong to metropolitan area and industrial and commercial towns.

The female respondents were the majority (83.3%, *n* = 648). The majority of respondents in this study were women, which may be due to the fact that most of the primary caregivers of young children are women, so women have more frequent contact with young children [50]. Additionally, they may be more able to respond more correctly to children’s participation in housework. About 56.1% (*n* = 434) of parents were aged 31–40, 37.4% (*n* = 289) were aged 41–50, 4.8% (*n* = 38) were aged under 30, and finally, 1.7% (*n* = 14) of parents were over 51 years old. In terms of parental education, most people had a college degree (63.4%, *n* = 491); the second highest was a master’s degree or above (20.5%, 159); the last was below high school (16.1%, *n* = 125).

### 2.2. Instrument

In this study, we developed a new “Young Children’s Participation in Housework” (YCPH) scale, since the scales of housework designed for children who are able to read and do housework independently in Taiwan [24] are not suitable for children ages 3–6. When developing the new scale, we referenced Lee and Tang [44] classification of young children’s housework and conducted a pilot study interviewing 15 preschool teachers, 17 parents, and 18 young children aged 4–6 to understand the items of children’s participations in those household tasks.

In the first draft of the questionnaire, the results of the pilot study provided important information on the content of scale for which household chores young children participated.

Although Dunn [51] designed the scale of Children Helping Out: Responsibilities, Expectations, and Support (CHORES) for school-aged children, which is not completely suitable for this study, the CHORES scale includes two subscales, self-care and family-care, which can clearly show the different characteristics of children’s participation in housework. Self-care chores are the tasks related to children‘s self-maintenance (e.g., put toys and picture books back in place), while family-care chores are the tasks related to family maintenance (e.g., set and clear table). Therefore, young children’s participation in housework designed in this study also consists of two parts: self-care chores and family-care chores. Since it is difficult for young children to do all the chores on their own, the response format of CHORES’s assistance scale can get a better idea of child’s involvement in chores by knowing if they can do it independently or need adult help. Therefore, the response format of the scale for the first draft, based on the design of CHORES, is a 7-point Likert scale, ranging from 0 (child is not expected to perform task) to 6 (child performs task on his/her own).

After completing the design of the first draft of the questionnaire, we invited 10 experts and scholars to check the validity of the YCHP scale and conducted a pre-test to analyze the reliability of the scale. A total of 113 pre-test questionnaires were distributed in this study, and 106 valid questionnaires were collected. Finally, the items of self-care chores scale were changed from 12 to 8, including putting back toys and books, putting dirty clothes in the hamper, getting school supplies ready, sorting things children brought home from preschool, folding their own clothes, making their own beds, washing cups, kettles, and lunch boxes, and arranging shoes neatly. The items of the housework scale have been changed from 21 to 16, including putting things back in place, setting the table, serving food, clearing the table, washing dishes, washing fruit, emptying the wastebasket, sorting garbage, putting family’s clothes in the hamper, hanging wet clothes, sweeping floors, cleaning bathrooms, carrying groceries, answering phone calls or doorbells, dusting doors and windows, and caring for younger siblings or other family members.

The response format of the YCHP scale was changed to a 5-point Likert scale, ranging from 1 (child is not expected to perform task) to 5 (child performs task on his/her own), since respondents in the pre-test indicated the 7-point Likert scale was too complex and time-consuming. In the reliability analysis, the Cronbach’s alpha internal consistency coefficient of the self-care chores subcale was 0.822, while the family-care chores subscale was 0.876. Both the self-care and family-care chores subscales showed evidence of strong internal consistency.

### 2.3. Measurements

This study used the self-compiled questionnaire as the source of research analysis. The measurement contents are described as follows.

#### 2.3.1. Outcome Variable—Children’s Housework Participation

Self-care chores related to children’s self-maintenance were measured by the sum of 8 items on the self-care chores subscale, while family-care chores related to family-maintenance were measured by the sum of 16 items on the family-care chores subscale. The respondents were rated on a scale of 1 (unable to do) to 5 (do it independently), with higher scores indicating that the child had more ability to do self-care chores or family-care chores. At the same time, the total of young children’s participation in housework was measured by the sum of the scores of two subscales, so as to understand the situation of young children’s participation in the overall young housework. The average scores of self-care and family-care housework for young children were 3.44 (SD = 0.80) and 2.91 (SD = 0.79).

#### 2.3.2. Independent Variable—Child’s Health

A child’s health was measured using the following questions: “How would you rate your child’s health?” Responses were rated on a scale of 1 (very poor) to 5 (very good), with higher scores indicating that the child was in better health. The average score of children’s health was 4.463 (SD = 0.606), indicating the majority of the children were healthy.

#### 2.3.3. Mediation Variables—Parents; Attitude toward Child’s Housework Participation, and Child’s Housework Preference

Two mediating variables are parents‘ attitude toward child’s housework participation and child’s housework preference. A parent’s attitude toward child’s housework participation was measured using the following questions: “Do you think it is important for children ages 3–6 to participate in housework?” Responses rated on a scale of 1 (very unimportant) to 5 (very important), with higher scores indicating that parents consider it more important for children to do chores. The average value of parents’ attitude was 4.337 (SD = 0.606), indicating that parents considered chores involvment as an important task for young children.

Children‘s housework preference was measured using the following questions: “Based on your observations, how much does this child like to do housework?” Since this question was to identify the direction (like or dislike) of children’s housework preference, the answer options were designed to have no intermediate options, only two directions: tending to like and tending to dislike. Responses were rated on a scale of 1 (dislike very much) to 4 (like it very much), and the higher the score, the more the child likes to do housework. The average score of child’s housework preference was 2.858 (SD = 0.699), indicating that young child slightly liked doing chores.

#### 2.3.4. Control Variables

The control variables in this study were: child’s gender, child’s sibling, and parents’ housework division. Child’s gender was measured using the following questions: child’s gender is: (1) male; (2) female. The numbers of male (50.5%, *n* = 391) and female children (49.5%, *n* = 348) were approximately equal. The child with sibling(s) or not was measured using the questions: “Does the child have siblings?” (1) No (2) Yes. Parents’ housework division was measured using the questions: “What is the status of the division of housework in your home?” About 32.0% (*n* = 248) of young children had siblings. Parental housework division was measured using the following questions: “What is the status of the division of housework in your home?” Responses were rated on a scale of 1 (all done by myself) to 5 (all done by spouse), with higher scores indicating that the spouse did more chores at home. The average score of parental housework division was 2.55 (SD = 0.73), indicating that respondents did slightly more housework than their spouse.

## 3. Results

### 3.1. Young Children’s Participation in Housework

This paper would like to illustrate the young children’s housework participation in Northern Taiwan before analyzing the mediating effect of the related variants. In terms of children’s participation in self-care housework, all but one housework item (folding their own clothes) differed significantly from the median score of 3. Children could complete half or more of each chore on their own, including putting back toys and books, putting dirty clothes in the hamper, getting school supplies ready, sorting things children brought home from preschool, and arranging shoes neatly, while they were not able to finish half or less chores on their own, including making their own beds and washing cups, kettles, and lunch boxes. Among them, the average scores of arranging shoes neatly (M = 4.66, SD = 0.75), putting dirty clothes in the hamper (M = 4.41, SD = 1.03), and putting back toys and books (M = 4.08, SD = 0.94) were higher than other items, indicating many children can complete, or complete most of, these housework items alone, while the average score of washing cups, kettles, and lunch boxes (M = 2.26, SD = 2.26) was the lowest, indicating that most children can only complete a small part or less than half of it on their own. Overall, the average score of children’s self-care housework was 3.44 (SD = 0.80), which means that most children can complete half or more of it on their own (Table 1).

In terms of children’s participation in family-care housework, 16 items composed this: putting things back in place, setting the table, serving food, clearing the table, washing dishes, washing fruit, emptying the wastebasket, sorting garbage, putting family’s clothes in the hamper, hanging wet clothes, sweeping floors, cleaning bathrooms, carrying groceries, answering phone calls or doorbells, dusting doors and windows, and caring for younger siblings or other family members. All but one housework item (setting the table) differed significantly from the median score 3.

Among them, the average scores of emptying the wastebasket (M = 4.71, SD = 0.73) and carrying groceries (M = 4.01, SD = 1.10) were higher than other items, indicating many children can complete, or complete most of, these housework items alone, while the average score of cleaning bathrooms (M = 1.53, SD = 0.93) was the lowest, indicating that most children were incapable of doing it or can only complete a small part of it on their own. Overall, the average score of children’s self-care housework was 2.91 (SD = 0.79), which means that most children can complete half of it on their own (Table 2). In addition, children’s ability to do self-care chores was higher than that of families (*t* = 15.51, *p* < 0.001).

### 3.2. Mediation Analysis

To determine the serial multiple mediation of parents’ attitude and child housework preference in the relationship between child’s health and household chores, the regression-based approach, PROCESS Model 6, as recommended by Hayes [52,53] was used. Obtained findings are presented from Figure 1 to Figure 2. The symbol of B is the unstandardized beta. B value represents the regression coefficients between two variables. The *c* value represents the total effect of the model, while the c’ value (unstandardized coefficient) presents the direct effect between independent variable and dependent variable. R^2^ can be interpreted as the percent of variance in the dependent variable that is explained by the set of predictor variables.

The serial multiple mediation of parents’ attitude and child’s housework preference in the relationship between child’s health and self-care chores, as can be seen in Figure 1, is as follows. Total effect (*c* = 1.689, CI = 0.970 to 2.407, *t* = 4.614, *p* < 0.001) of child’s health on self-care chores was at a significant level (Step 1). In addition, the direct effects of child’s health on both the parent’s attitude (B = 0.181, CI = 0.110 to 0.252, *t* = 5.014, *p* < 0.001) and child’s housework preference (B = 0.087, CI = 0.007 to 0.166, *t* = 2.140, *p* < 0.05) were at significant levels. The direct effect of parent’s attitude as the first mediating variable on the second mediating variable of child’s housework preference (B = 0.205, CI = 0.127 to 0.282, *t* = 5.153, *p* < 0.001) was on a significant level (Step 2). A review of the direct effects of mediating variables on self-care housework, on the other hand, showed that the effects of parent’s attitude (B = 1.161, CI = 0.460 to 1.862, *t* = 3.253, *p* < 0.001) and child’s housework preference (B = 2.105, CI = 1.480 to 2.731, *t* = 6.604, *p* < 0.001) were at significant levels (Step 3). When child’s health and all other mediating variables were simultaneously entered into the equation (Step 4), the relationship between the child’s health and self-care housework, in relation to the direct effect, was at a significant level (c’ = 1.218, CI = 0.514 to 1.922, *t* = 3.394, *p* < 0.001). Based on this result, the mediating variables were observed to partially mediate between the child’s health and self-care housework. In addition, the model overall was seen to be at a significant level (F (6–768) = 21.167, *p* < 0.001) and explained 14.2% of the total variance in self-care housework (Figure 1, Table 3).

The results of indirect effects and specific effects of child’s health on self-care housework through parents’ attitude and child’s housework preference were included in Table 2. As seen in Table 2, the total indirect effect (the difference between total and direct effects/c-c’) of child’s health through parent’s attitude and child’s housework preference on self-care housework was statistically significant (effect = 0.078, CI = 0.035 to 0.134). Within the tested model, when considering the mediating variables separately and together, in relation to the mediating indirect effects of the child’s health on self-care housework, single mediation of parent’s attitude (effect = 0.210, CI = 0.073 to 0.393) was found statistically significant, and single mediation of child’s housework preference (effect = 0.182; CI = 0.017 to 0.367) was found statistically significant.

The serial multiple mediation of parents’ attitude and child’s housework preference in the relationship between child’s health and family-care housework, as can be seen in Figure 2, total effect (c = 3.516, CI = 2.100 to 4.932, *t* = 4.875, *p* < 0.001) of child’s health on family-care housework was at a significant level (Step 1). In addition, the direct effects of the child’s health on both the parent’s attitude (effect = 0.181, CI = 0.110 to 0.252, *t* = 5.014, *p* < 0.001) and child’s housework preference (B = 0.087, CI = 0.007 to 0.166, *t* = 2.140, *p* < 0.05) were at significant levels. The direct effect of the parents’ attitude as the first mediating variable on the second mediating variable of child’s housework preference (B = 0.205, CI = 0.127 to 0.282, *t* = 5.153, *p* < 0.001) was on a significant level (Step 2). A review of the direct effects of the mediating variables on family-care housework, on the other hand, showed that the effects of the parents’ attitude (B = 3.016, CI = 1.632 to 4.399, *t* = 4.280, *p* < 0.001) and child’s housework preference (B = 3.548, CI = 2.312 to 4.783, *t* = 5.637, *p* < 0.001) were at significant levels (Step 3). When a child’s health and all other mediating variables were simultaneously entered into the equation (Step 4), the relationship between the child’s health and family-care housework, in relation to the direct effect, was at a significant level (c’ = 2.531, CI = 1.140 to 3.922, *t* = 3.572, *p* < 0.001). Based on this result, the mediating variables were observed to partially mediate between the child’s health and family-care housework. Additionally, the model overall was seen to be at a significant level (F (6–768) = 21.345, *p* < 0.001) and explained 14.3% of the total variance in family-care housework (Figure 2, Table 4).

The results of the indirect effects and specific effects of the child’s health on family-care housework through PACHP and child’s housework preference were included in Table 4. As seen in Table 4, the total indirect effect (the difference between total and direct effects/c-c’) of the child’s health through PACHP and the child’s housework preference on family-care housework was statistically significant (effect = 0.132, CI = 0.058 to 0.233). Within the tested model, when considering the mediating variables separately and together, in relation to the mediating indirect effects of child’s health on family-care housework, single mediation of PACHP (effect = 0.547, CI = 0.241 to 0.932) was found to be statistically significant, and single mediation of child’s housework preference (effect = 0.307; CI = 0.027 to 0.647) was found to be statistically significant.

## 4. Discussion

### 4.1. Young Children’s Ability to Participate in Self-Care Housework Is Slightly Better than Family-Care Housework

Although parents believe that it is important for children to participate in housework, because participation in housework will help to adapt to future life [27,34], the finding did not support the parents’ expectations. Research has also found that young children’s ability to participate in different chores varies widely, and of the 24 children’s chores, only four can be performed almost independently, including arranging shoes neatly, putting dirty clothes in the hamper, putting back toys and books, putting dirty clothes in the hamper, and carrying groceries. The varying levels of difficulty of these chores [44] and the immaturity of young children’s motor, cognitive, and language development are reasons why they will not be able to perform all of the chores alone [54]. Lee [55] also explained that children need to learn housework in line with their physical and cognitive development. In addition, this study also found that young children performed better in self-care chores than in family-care chores. Goodnow [56] also pointed out that, as children grow older, they will gradually shift from focusing on household chores related to themselves to family chores that meet the needs of other family members. Additionally, the results of this study seem to echo Goodnow’s argument.

### 4.2. The Effect of Children’s Health on Children’s Participate in Housework Is Partly through Parents’ Attitudes and Children’s Preferences

Although this study confirms the past findings that children’s health [27,30], parents’ attitudes [25,57], and children’s preferences [37,38] all have impact on children’s participation in housework, it is more important to point out that the influences on children’s participation in housework should be analyzed from a more precise perspective. That is, it is necessary to further understand the relationship among the variables that influence children’s participation in housework.

This study confirms that, when children are unhealthy, their participation in both self-care and family-care housework will decrease by changing their parents’ attitude towards children’s housework. This is mainly because parents are the arrangers of their children’s housework [42], so when they pay more attention to children’s health, instead of children’s housework, parents will naturally not arrange for sick children to do housework. At the same time, this study also found that, when young children are in poor health, this creates lower housework preferences, which, in turn, makes them less able to participate in housework. This may be a result of physical insufficiency and physical defects caused by illness, making it difficult for sick children to do housework, which, in turn, makes sick children less willing to do housework [58]. Of course, if a child is sick, parents often feel that the child is incapable of performing these housework [59], so they give the child less encouragement and guidance on housework, which may make the child less interested in doing housework, thereby decreasing housework participation [43].

## 5. Conclusions

Although previous studies have found that young children like to participate in adults’ housework, such as vacuuming and tidying up with their parents [60], this study found that young children’s participation in housework is limited. The development of housework skills takes time, but if parents are impatient and unable to let their children learn gradually, according to their own developmental time, it will affect their children’s housework skills [61]. It is suggested that adults should use the stage when children like to actively assist in housework to teach children to do housework and let children have the right to choose to participate in housework, so as to strengthen their interest in doing housework [55] and learn more housework skills.

This study confirms that, when analyzing the factors of children’s housework participation, it is necessary to have a clearer understanding of the relationship between the variables, in order to further construct a more complete model framework that affects children’s housework participation. In addition, this study is limited to families where young children live with their parents, but past studies have found that children do more housework in dual-income families and single-parent families [9,21] and do less housework in three-generation families [62]. For younger children, whether their participation in housework varies by family type is another direction that needs to be further explored in future research. Finally, whether the sense of housework fairness concerned by adults starts from the housework participation in early childhood, and whether the participation of young children in housework is also related to culture, these issues are all directions that future research can explore.

One of the limitations of this study is that, because the language and cognitive abilities of young children are not yet mature, the questionnaire cannot be answered directly by young children, but by parents who observe their children’s participation in housework at home. Therefore, there may be some social expectation bias when answering. In addition, the survey area of this study includes three counties and cities in northern Taiwan, so the results of the study will be limited in estimating other parts of Taiwan. In particular, those places belong to the urban area, so the items of young children’ housework may be different from those in rural areas, such as assisting in feeding chickens, simply planting vegetables, and other household chores that are only available in rural areas. Future research can further analyze the possible impact of urban–rural regional differences on children’s housework participation.

In the model of this study, the presence of siblings was considered as a control variable. Previous studies have found that children with older siblings are more likely to imitate what siblings do [27], so when young children have older siblings, they tend to do more housework [47]. Therefore, future research can also consider including the number of siblings, having older or younger siblings, etc., into the analysis.

## Figures and Tables

**Figure 1 ijerph-19-15448-f001:**
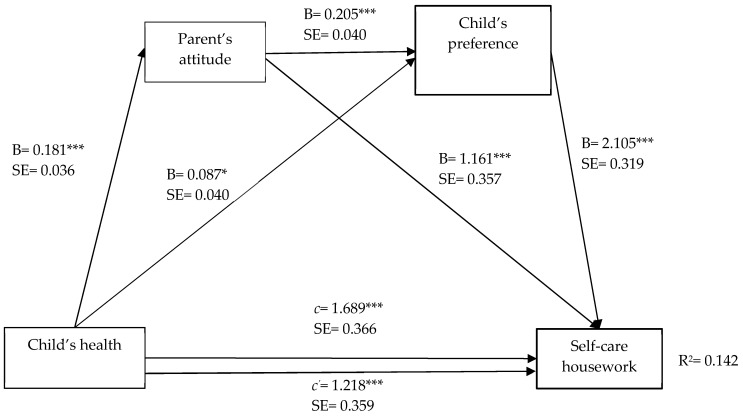
Serial multiple mediation of parents’ attitude and child’s housework preference in the relationship between child’s health and self-care housework, with non-standardized beta values. *** *p* < 0.001, * *p* < 0.05.

**Figure 2 ijerph-19-15448-f002:**
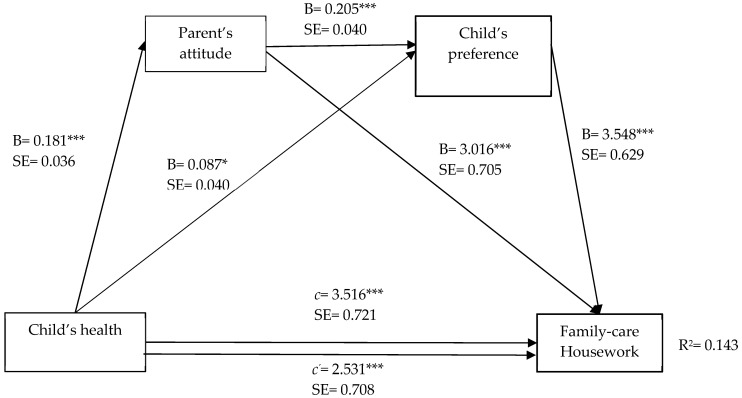
Serial multiple mediation of PACHP and Child’s housework preference in the relationship between Child’s health and family-care housework, with non-standardized beta values. *** *p* < 0.001, * *p* < 0.05.

**Table 1 ijerph-19-15448-t001:** Description of self-care housework (N = 775).

Items	*Mean*	*SD*	*t*
putting back toys and books	4.08	0.95	31.96 ***
putting dirty clothes in the hamper	4.41	1.03	38.20 ***
getting school supplies ready	3.17	1.26	3.76 ***
sorting things children brought home from preschool	3.22	1.34	4.52 ***
folding their own clothes	2.94	1.32	−0.136
make their own beds	2.82	1.42	−3.54 ***
washing cups, kettles, and lunch boxes	2.26	1.29	−16.05 ***
arranging shoes neatly	4.66	0.74	61.33 ***
self-care housework	3.44	0.80	15.51 ***

*** *p* < 0.001.

**Table 2 ijerph-19-15448-t002:** Descriptive statistics of family-care housework (N = 775).

Items	Mean	SD	*t*
putting things back in place	3.93	1.04	24.90 ***
setting the table	3.04	2.39	0.70
serving food	2.68	1.48	−6.07 ***
clearing the table	2.87	1.37	−2.60 **
washing dishes	2.08	1.30	−19.66 ***
washing fruit	2.32	1.40	−13.50 ***
emptying the wastebasket	4.71	0.73	65.11 ***
sorting garbage	3.23	1.52	4.12 ***
putting family’s clothes in the hamper	3.74	1.58	12.98 ***
hanging wet clothes	2.05	1.31	−20.15 ***
sweeping floors	2.53	1.30	−9.94 ***
cleaning bathrooms	1.53	0.93	−44.13 ***
carrying groceries	4.01	1.10	25.39 ***
answering phone calls or doorbells	3.15	1.60	2.55 *
dusting doors and windows	2.19	1.36	−16.45 ***
caring for younger siblings or other family members	2.55	1.36	−9.22 ***
**family-care housework**	2.91	0.79	−3.08 **

* *p* < 0.05; ** *p* < 0.01; *** *p* < 0.001.

**Table 3 ijerph-19-15448-t003:** Serial mediation effects on the relationship between Child’s health and Self-care Chores.

Model Pathways	B	*SE*	*t*	*p*	*LL 95%CI*	*UL95%CI*
Child’s health -> Parent’s attitude	0.181	0.036	5.014	<0.001 ***	0.110	0.252
Child’s health -> Child’s preference	0.087	0.040	2.140	<0.05 *	0.007	0.166
Parent’s attitude -> Self-care chores	1.161	0.357	3.253	<0.001 ***	0.460	1.862
Child’s preference -> Self-care chores	2.105	0.319	6.604	<0.001 ***	1.480	2.731
Parent’s attitude -> Child’s preference	0.205	0.040	5.153	<0.001 ***	0.127	0.282
	*effect*	*SE*	*t*	*p*	*LL 95%CI*	*UL95%CI*
Total model effect	1.689	0.366	4.614	<0.001 ***	0.970	2.407
Direct effect	1.218	0.359	3.394	<0.001 ***	0.514	1.922
Total indirect effect	0.471	0.128			0.232	0.734
Child’s health -> Parent’s attitude -> Self-care chores (model 1)	0.210	0.083			0.073	0.393
Child’s health -> Child’s preference -> Self-care chores (model 2)	0.182	0.089			0.017	0.367
Child’s health -> Parent’s attitude -> Child’s preference -> Self-care chores (model 3)	0.078	0.025			0.035	0.134

Note. Models include controls for child’s gender, siblings, and parents’ housework division. All pathways are unstandardized. Indirect effects were computed using 5000 bootstrap samples. Unstandardized indirect effects are shown outside parentheses. Standardized indirect effects are shown inside parentheses. *** *p* < 0.001, * *p* < 0.05.

**Table 4 ijerph-19-15448-t004:** Serial mediation effects on the relationship between Child’s health and Family-care housework.

Model Pathways	B	*SE*	*t*	*p*	*LL 95%CI*	*UL95%CI*
Child’s health -> Parent’s attitude	0.181	0.036	5.014	<0.001 ***	0.110	0.252
Child’s health -> Child’s housework preference	0.087	0.040	2.140	<0.05 *	0.007	0.166
Parent’s attitude -> Family-care housework	3.016	0.705	4.280	<0.001 ***	1.632	4.399
Child’s preference -> Family-care housework	3.548	0.629	5.637	<0.001 ***	2.312	4.783
Parent’s attitude -> Child’s preference	0.205	0.040	5.153	<0.001 ***	0.127	0.282
	*effect*	*SE*	*t*	*p*	*LL 95%CI*	*UL95%CI*
Total model effect	3.516	0.721	4.875	<0.001 ***	2.100	4.932
Direct effect	2.531	0.078	3.572	<0.001 ***	1.140	3.922
Total indirect effect	0.985	0.247			0.539	1.511
Child’s health -> Parent’s attitude -> Family-care housework (model 1)	0.547	0.180			0.241	0.932
Child’s health -> Child’s preference -> Family-care housework (model 2)	0.307	0.159			0.027	0.647
Child’s health -> Parent’s attitude -> Child’s preference -> Family-care housework (model 3)	0.132	0.045			0.058	0.233

Note. Models include controls for child’s gender, siblings, and parents’ housework division. All pathways are unstandardized. Indirect effects were computed using 5000 bootstrap samples. Unstandardized indirect effects are shown outside parentheses. Standardized indirect effects are shown inside parentheses. *** *p* < 0.001, * *p* < 0.05.

## Data Availability

The data presented in this study are available on request from the first author.

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
