# Peer review of "Young Children’s Housework Participation in Taiwan: Serial Multiple Mediations"

_ijerph, 2022, doi:10.3390/ijerph192315448_

Round 1

Reviewer 1 Report

The article's subject is interesting and well-exposed. The references support the hypothesis of the author. At the same time, there are some problems with the definition of the topics. In particular related to the idea that is common or socially accepted that very young children were involved in the housework. These topics introduce the necessity to introduce the cultural issue, respect to the country interested, and respect to the families involved.  At the same time is necessary to introduce the role played by the territorial dimension. Big city, countryside, etc.,  which differences and why. All of these are fundamental to avoiding a self-explication of the phenomenon.

Reviewer 2 Report

The article was written in a proper way, but there are still some minor corrections to be made. Lines 112-122 needlessly repeat what appears on lines 125-134. It is worth explaining how the household chores list was built. Why were these elements chosen? On line 211 there is a description of a scale with a value from 1 to 4. Explain why this scale was used when other questions are rated on a scale from 1 to 5. The control variable related to having siblings is imprecise. It is worth showing the limitation in the form of not including information on the number of siblings, having older or younger siblings, etc. In line 272, figure 3 was mentioned, but the article lacks such a figure. It is also worth precisely explaining all the symbols on the occasion when their are used for the first time in the text or in the description of a table (e.g. B etc.).

Reviewer 3 Report

The study examined children participation in household chores and parents attitude. It also examined parents attitude and child preference on childrens health. The introduction, methods, and analyses is an interesting contribution to literature. The method is explicit for replication by other scholars.

Is there any reason(s) why we have more female respondents than male? The margin is wide

Round 2

Reviewer 1 Report

After the revisions, the article is ready for publication.